# The Synergistic Effect of Biochar-Combined Activated Phosphate Rock Treatments in Typical Vegetables in Tropical Sandy Soil: Results from Nutrition Supply and the Immobilization of Toxic Metals

**DOI:** 10.3390/ijerph19116431

**Published:** 2022-05-25

**Authors:** Zhiwei Zhang, Beibei Liu, Zhenli He, Pan Pan, Lin Wu, Bigui Lin, Qinfen Li, Xinchun Zhang, Zhikang Wang

**Affiliations:** 1College of Eco-Environmental Engineering, Guizhou Minzu University, Guiyang 550025, China; billlaoz909@gmail.com; 2Hainan Key Laboratory of Tropical Eco-Circular Agriculture, Hainan Engineering Research Center for Non-Point Source and Heavy Metal Pollution Control, Institute of Environmental and Plant Protection, Chinese Academy of Tropical Agricultural Sciences, Haikou 570100, China; ppan2528@163.com (P.P.); wu_lin88@163.com (L.W.); linbigui@163.com (B.L.); qinfenli2005@163.com (Q.L.); xinxin7238@163.com (X.Z.); 3Danzhou Scientific Observing and Experimental Station of Agro-Environment, Ministry of Agriculture and Rural Affairs, P.R. China, National Agricultural Experimental Station for Agricultural Environment, Danzhou 571700, China; 4Indian River Research and Education Center, Institute of Food and Agricultural Sciences, University of Florida, Fort Pierce, FL 32611, USA; zhe@ufl.edu

**Keywords:** composite amendment, activated phosphate rock, biochar, tropical soil, vegetable

## Abstract

Sandy soils in tropical areas are more vulnerable to potential toxic elements as a result of their low nutrition. The composite addition of biochar and phosphate material is considered a promising method of immobilizing toxic metals in sandy soils, but the synergistic effects of this process still need to be further explored, especially in typical tropical vegetables. In this study, a pot experiment was conducted to evaluate the agronomic and toxic metal-immobilization effects of single amendments (phosphate rock, activated phosphate rock, and biochar) and combined amendments, including biochar mixed with phosphate rock (BCPR) and biochar mixed with activated phosphate rock (BCAPR), on vegetables grown in tropical sandy soil. Among these amendments, the composite amendment BCAPR was the most effective for increasing Ca, Mg, and P uptake based on water spinach (*Ipomoea aquatica* L.) and pepper (*Capsicum annuum* L.), showing increased ratios of 22.5%, 146.0%, and 136.0%, respectively. The SEM-EDS and FTIR analysis verified that the activation process induced by humic acid resulted in the complexation and chelation of the elements P, Ca, and Mg into bioavailable forms. Furthermore, the retention of available nutrition elements was enhanced due to the strong adsorption capacity of the biochar. In terms of cadmium (Cd) and lead (Pb) passivation, the formation of insoluble mineral precipitates reduced the mobility of these metals within the BCAPR treatments, with the maximum level of extractable Cd (86.6%) and Pb (39.2%) reduction being observed in the tropical sandy soil. These results explore the use of sustainable novel cost-effective and highly efficient bi-functional mineral-based soil amendments for metal passivation and plant protection.

## 1. Introduction

Sandy soil covers an area of over 900 million hectares around the world and plays an undeniable role in the production of food crops in the subhumid tropics [1]. Given the low availability of water, soil organic matter (SOM), and nutrients, tropical sandy soils are typically infertile. Sandy soil degradation, which includes SOM loss, acidification/alkalization, and erosion, might further reduce crop productivity [2]. Specifically, acidity-related toxicity and the low nutrient availability of acidic soils are major limiting factors for vegetable growth. The high iron and aluminum oxide levels in tropical sandy soils result in them having a high phosphorus (P) adsorption capacity, and thus, the amounts of P that are extracted from harvested products is much less than the amounts that are added via fertilizers [3,4]. As an exception to their low nutrient retention capacity, sandy soils are more vulnerable to potential pollution by toxic elements [5,6]. To increase the nutrition of and the ability to immobilize potential toxic elements in crops grown in tropical sandy soil, it is desperately necessary to improve soil amendments to achieve desirable soil quality.

Phosphate rock has been reported to be as effective as superphosphate when used as a soil amendment, and it is more cost-effective for correcting P deficiencies [7]. The exploitation and utilization of low grade phosphate rock resources have resulted in them having low utilization efficiency, and a large number of researchers have chosen phosphate rock to manage the heavy metal pollution. To improve the efficiency of the nutritional and environmental functions of phosphate rock, organic molecules including oxalic acid, humic acid, and sodium lignosulfonate were used to activate phosphate rock to fabricate activated phosphate rock. Compared to phosphate rock, activated phosphate rock has been confirmed to be more effective for heavy metal (e.g., Pb, Zn, and Cd) immobilization and for the P supply [8,9,10,11,12,13,14,15,16]. Moreover, the above activation process has also been identified being able to release less labile heavy metals compared to traditional manufacturing methods, such as acidification with concentrated sulfuric acid [17,18]. Shao et al. used sodium lignosulfonate and humic acid to fabricate two kinds of activated phosphate rock. The results showed that the Cd that was adsorbed on the activated phosphate rock was achieved through ligand exchange and the formation of internal complexes, which significantly decreased the soil-exchangeable Cd by 48.9–55.0% [19].

As a widely used soil amendment, biochar has been extensively applied to improve the physicochemical properties of soil, such as the porosity, cation exchange capacity, and pH, and subsequently, the soil’s holding capacity for water and nutrients and crop production [20,21,22,23]. For acidic sandy soils, biochar improved the soil organic carbon pool, water-holding capacity, the nutrient supply capacity and adsorption through increasing the cation exchange capacity [3]. The combination of biochar and phosphate material has been recognized as a desirable soil amendment for modifying soil properties and crop growth, especially in acidic, P-fixing, and P-deficient soils. Biochar-based phosphate fertilizers generated via a co-pyrolysis process using biomass and soluble phosphates or phosphate rock have been developed, and some studies have introduced bacterial strains to promote the solubilization of phosphate rock to enhance P utilization [24]. Given that they were able to improve biochar carbon retention, slow nutrient release and stabilize heavy metals in soil, it has been proven that the agronomic effectiveness of biochar-based phosphate fertilizers is better than that of soluble phosphate fertilizers [25,26]. Recently, the effectiveness of simultaneous applications of biochar and phosphate rock/P fertilizer were investigated, and these investigations showed that biochar provided additional soluble P and supplied adsorption sites for phosphate, preventing the evolution of an unavailable form of P [3]. The addition of biochar and P fertilizer exhibited positive effects on the growth and nutrient uptake of *L. multiflorum* in acidic Cd-contaminated soils [27].

Our previous study showed that the application of humic acid-activated dolomite phosphate rock and biochar were effective amendments for immobilizing heavy metals in contaminated sandy soils, and the humic acid-activated dolomite phosphate rock reduced the amount of extractable Cd^2+^ and Pb^2+^ by 87.2 and 76.0% in Alfisol and by 91.3 and 76.3% in Spodosol soil, respectively, compared to the control [28]. Biochar’s nutrient supply and adsorption capacities are due to its large surface area, highly porous structure and strong ion exchange capacity, and because of this, the combination of activated phosphate rock and biochar may exhibit an excellent effect not only for toxic metal passivation but also for the ability of vegetables to uptake nutrients. However, to date, the synergistic effects of biochar combined with activated phosphate rock (BCAPR) on vegetables in tropical sandy soil are still unclear. To address this issue, the agronomic effectiveness as well as the effectives of single amendments (phosphate rock, activated phosphate rock and biochar) and combined amendments such as biochar combined with phosphate rock (BCPR) and biochar combined with activated phosphate rock (BCAPR) to immobilize toxic metals in heavy metal-contaminated tropical sandy soil and vegetable production were investigated in the present study. Sandy soil covers a large area in Hainan province, covering a total area of 0.32 million hectares. It accounts for about 9% of the island’s total area, and nearly 70% of the sandy soil in Hainan province is cultivated. Thus, the sandy soil comprising the cultivated land in Hainan province was selected for this study.

## 2. Materials and Methods

### 2.1. Study Area, Soil, and Materials

The sandy soil was collected at a depth of 0–20 cm at three parallel locations in Dongfang, Hainan province, China (N19°31′22.43″, E109°34′36.02″), which is located at the northern edge of the tropical region. The sampling sites were located on cultivated land, and in the area, the soil pH is 5.61. The annual average precipitation and annual average temperature of the experimental site are 1815 mm and 23.5 °C, respectively. The collected soil samples were air-dried and ground to pass through a 2 mm sieve prior to analysis for chemical properties and use. The soil samples were sandy, containing 92% sand, 3% silt, and 5% clay. Phosphate rock was purchased from Beiqi Chemical Inc. in Huibei province, China. Biochar was derived from coconut husks (pyrolyzed at a temperature of 650 °C) and supplied by Dongjiao Coconut Activated Carbon Inc in Hainan province of China. The detailed information about the biochar and phosphate rock are presented in the Appendix A.

### 2.2. Preparation and Characterization of Amendments

To fabricate the activated phosphate rock, phosphate rock samples were ground to less than 100 mesh and then activated by humic acid as per the previous study [28]. Briefly, the phosphate rock samples were thoroughly mixed with humic acid (5%), and then deionized water (15% moisture) was added. Afterwards, the mixture was manually ground with a pestle in the mortar for 15 min. Before analyzing the chemical properties, the mixtures were air dried and ground to pass through a 1 mm sieve. The combined amendment BCPR was fabricated by mixing biochar and phosphate rock at a weight ratio of 1:1, and BCAPR was fabricated by mixing biochar and activated phosphate rock at the same weight ratio. The following single and composite amendments were used in the study: phosphate rock, activated phosphate rock, biochar, BCPR, and BCAPR. These amendments were prepared for the further analysis and for use in the pot experiment. The morphology and energy dispersive spectra of each amendment was determined using SEM-EDS (TM40000 Plus, Hitachi, Tokyo, Japan), which was conducted by following the method described in the previous study [19]. The functional groups of each amendment were analyzed by means of Fourier-transform infrared (FTIR) spectroscopy using a Perkin Elmer Frontier spectrometer. The FTIR spectra of the ground samples were collected over an average of 32 scans with a resolution of 4 cm^−1^ and a range of 4000 to 400 cm^−1^.

### 2.3. Pot Experiment

The pot experiment was conducted using the collected soil samples and was carried out under greenhouse conditions in Hainan, China (N20°02′46.78″, E110°11′44.14″). Six types of treatments were set up with three replicates as follows: the control (without amendments) treatment, the phosphate rock treatment, the activated phosphate rock treatment, the biochar treatment, the BCPR treatment, and the BCAPR treatment. For each treatment except for the control, the amendment was applied to the soil at a rate of 1% and thoroughly mixed with the soil before being transferred into pots (diameter: 20 cm, height: 20 cm, bulk density of the soil: 1.56 g cm^−^^3^). To avoid any possible negative impact caused by the amendment, such as the effects of volatile substances on plant growth, the mixture of soil and amendment was maintained at a 60% soil water holding capacity for four weeks. Afterwards, water spinach (*Ipomoea aquatica* L.) and pepper (*Capsicum annuum* L.) were cultivated in sequence aiming to evaluate the agronomic performances of different amendments in the short and medium terms respectively. Water spinach was grown for 30 days, being succeeded by pepper for further 60 days. Detailed information can be found in the Appendix A.

Soluble P fertilizer (NH_4_H_2_PO_4_) was used in the control treatments, and urea and KCl were applied in all of the treatments to reach a fertilizer level that is commonly used in agricultural production. To avoid differences in the growth conditions, the pots were placed randomly and their positions were changed weekly. After the designed growing period, the chlorophyll contents in the vegetable samples were determined using a SPAD-502 Meter following previously established methods [29,30]. The vegetable samples were harvested, washed with tap water and deionized water, and then dried at 70℃ in an oven. The lengths and weights of the plants were measured before they were pulverized. Finally, the sandy soil in each pot was taken out, air-dried, and sieved for the determination. To explore the passivation effect of the various amendments on the Cd and lead (Pb) contents in sandy soil, the adsorption capacity of sandy soil with various amendments on Cd and Pb was determined through isothermal adsorption experiments, as per the method in the Appendix A.

### 2.4. Analysis of Soil and Vegetable

The pH of the soil samples was measured using a pH meter (Model 220, Denver Instrument, Denver, CO, USA) following the U.S. Environmental Protection Agency method 150.1, that is, the pH of the soil sample was measured in deionized water at a solid/water ratio of 1:1. The organic matter in the soil sample was determined using a TOC analyzer by following the method described in a previous study [16]. The available nutrients and metals in the soil samples with different treatments and in the vegetable samples were measured by means of extraction from the samples with Mehlich 3 (M3) solution at a 1:10 solid to solution ratio [31]. Afterwards, the extracts were filtered through a 0.45-µm membrane. The subsamples of the filtrate were acidified and then analyzed to determine the concentrations of dissolved P, calcium (Ca), magnesium (Mg), Cd, and Pb. Each portion of the plant samples (0.4 g) was digested with 5 mL of concentrated HNO_3_/H_2_O_2_. Following a previously described method [18], the concentrations of P, Ca, Mg, Pb, and Cd in the digested samples were determined by using an ICP-MS (NexION 2000, PerkinElmer, Waltham, MA, USA).

### 2.5. Statistical Analysis

The mean and standard error of all of the samples including the replications were analyzed using the analysis of variance (ANOVA) by SPSS 10.0 statistical package (SPSS, Chicago, IL, USA). The data of the plant dry biomass, plant height, chlorophyll content, nutrient concentration, and metal concentration among the various treatments were tested by Kolmogorov–Smirnov test, and then the homogeneity of variance was measured. Afterwards, the differences in the plant dry biomass, plant height, chlorophyll content, nutrient concentration, and metal concentration among the various treatments were analyzed using one-way ANOVA. The significant differences among the means were determined at *p* < 0.05 and via Duncan’s multi-range tests. The Pearson correlation was performed to investigate the significance of the correlations.

## 3. Results

### 3.1. Characterization of Soil Amendment

The single and composite amendments were characterized by SEM-EDS (Figure 1) and FTIR (Figure 2). Based on the SEM images of the phosphate rock and activated phosphate rock amendments, it was easy to see that the original agglomerate crystal structures in the phosphate rock transformed and developed an aggregated porous structure via the activation process of humic acid. The biochar exhibited a rough porous structure and had an average pore diameter of approximately 0.5 mm (Appendix A). The phosphate rock and activated phosphate rock particles were roughly 0.05 mm in size and tended to be randomly distributed over the outer and inside surfaces of the biochar. The BCAPR amendment had high levels of C, P, Ca, and Mg, which were revealed by EDS analysis. FTIR analysis was performed to further investigate the combing form of the phosphorous in the different amendments. As shown in Figure 2, biochar was characterized by a shoulder at around 1582 cm^−1^, which can be attributed to C=O and C=C stretching of the aromatic rings. The reduction in the relative intensity of the signal in the above-mentioned area was investigated when the biochar was combined with P materials (phosphate rock and activated phosphate rock). In the FTIR spectrum of the phosphate rock amendment, the peaks at 1453 and 1437 cm^−1^ showed the antisymmetrical elastic vibrating ν3 absorption patterns of CO_3_^2−^, and the peak at 880 cm^−1^ showed an elastic vibrating ν2 absorption pattern for CO_3_^2−^. The peaks at 964 and 467 cm^−1^ in the phosphate rock amendment showed a symmetrical elastic vibrating ν1 absorption pattern for PO_4_^3-^ and an elastic vibrating ν2 absorption pattern for PO_4_^3−^, both of which were the result of the phosphate groups that are present in the apatite lattice of phosphate rock.

### 3.2. Effects of Amendments on Soil Characteristics

The impacts of the different amendments on the pH, soil organic matter, available nutrition concentration, and extractable Cd and Pb in the soil samples are presented in Table 1. For both the water spinach and pepper, the amendments containing biochar and phosphate rock induced a higher soil pH value than the other amendments. In detail, the application of activated phosphate rock alone was less effective than it was when combined with biochar, that is, the BCAPR treatment elicited the highest concentration of available P, increasing the available P by 1.88 times compared to the control. Additionally, the levels of available Ca and Mg were also improved in the amendment treatments containing phosphate rock and activated phosphate rock compared to the other amendments. During the first and second cropping cycles, the reduction in the maximum extractable Cd (86.6%) and Pb (39.2%) was observed in the BCAPR amendment treatments respectively. The adsorption results of Cd and Pb for the single and composite amendments in the sandy soil samples also supported the above conclusion (Appendix A).

### 3.3. Plant Growth, Biomass, and Nutrient Uptake

The plant height, the leaf chlorophyll content, and biomass of the water spinach (first crop) and pepper (second crop) planted in successive cultivation periods with different amendments are shown in Figure 3. In the first cropping cycle, the application of the different amendments had no significant influence on the dry biomass of the root and shoot, with the exception of the phosphate rock and the BCAPR treatment which presented a lower and higher shoot dry biomass respectively. During the second cropping cycle with the pepper plants, the dry biomass of the shoots increased by 11.2%, 2.95%, 15.3%, 5.11%, and 7.51%, and the dry biomass of the roots increased by 8.06%, 6.05%, 7.12%, 10.9%, and 13.5% after the application of the biochar, phosphate rock, activated phosphate rock, BCPR, and BCAPR amendments respectively, compared to the control. Both the water spinach and pepper showed significantly increased growth (*p* < 0.05) after the application of single and composite amendments compared to the control, and no significant difference was observed among different amendments for water spinach. However, for the pepper plants grown, the application of activated phosphate rock and BCAPR increased their height by 47.5% and 58.2% respectively, compared to the control. Furthermore, there were no significant differences among the various treatments when considering the leaf chlorophyll content of the pepper. For the water spinach, the activated phosphate rock treatments increased the leaf chlorophyll content by 7.95%, compared to the control.

The uptake of P, Ca, and Mg elements by the water spinach and pepper in the different treatments was presented in Figure 4. Generally, it is easy to see that the application of the activated phosphate rock and BCAPR had improved the P, Ca, and Mg uptake by the plants significantly. Unlike the uptake of P, Ca, and Mg by the water spinach observed for the different treatments, the application of phosphate rock, activated phosphate rock, BCPR and BCAPR significantly improved P, Ca, and Mg uptake by the pepper, compared to the control (*p* < 0.05). The pepper showed the maximum P, Ca, and Mg uptake after the application of BCAPR amendments compared to the other amendments. Specifically, the application of BCAPR improved P, Ca, and Mg uptake by 22.5%, 146%, and 136% respectively, compared to the control.

### 3.4. Effects of Amendments on Cd and Pb Uptake

As shown in Figure 5, the concentrations of Cd and Pb in the aerial parts of the water spinach and pepper in all the amendment treatments were significantly reduced compared to the control (*p* < 0.05), and differed among the five kinds of amendment treatments. There was no significant difference between the treatments with different amendments for Pb uptake, but a difference in the reduced Cd uptake was observed among the different amendment treatments. Compared to the control, the uptake of Cd by the water spinach reduced by 64.9% with the application of biochar, which was significantly lower than that of the water spinach grown with phosphate rock (78.6%) and activated phosphate rock amendments (80.9%). For the uptake of Cd by pepper, the application of BCAPR showed the highest reducing ratio (67.5%) compared to the control, and the application of BC showed the lowest reducing ratio (36.0%). Additionally, the Pb uptake by the water spinach and pepper reduced by 68.1–73.7% and 25.9–37.0%, respectively. Overall, the BCAPR amendment was the most effective for the toxic metal passivation in the tropical sandy soil sample than the other single and composite amendments.

## 4. Discussion

### 4.1. The Synergistic Mechanisms of Biochar and Activated Phosphate Rock on Agronomic Effects

The activation of dolomite phosphate rock with selected organic molecules resulted in a substantial increase in water-soluble P and the P amendment continuously supplied P to meet the growth requirements of maize (*Zea mays*) and millet (*Pennisetum glaucum*) [12]. In this study, activated phosphate rock fabricated by humic acid showed a higher nutrition supply for both water spinach and pepper compared to phosphate rock and activated phosphate rock. The formation of a Ca-P precipitate has a significant negative relationship with the humic acid concentration, and the activation process induced by humic acid was observed to release less labile toxic metals compared to traditional manufacturing methods such as acidification using concentrated sulfuric acid [17,18]. It could increase the release of P, Ca, and Mg in phosphate rocks due to the complexation and chelation of these elements into bioavailable forms. Our previous study proved that both the humic acid activated phosphate rock and the biochar had great potential to remedy the toxic levels of Cd and Pb in contaminated sandy soils, but their agronomic and nutrition supply effects on the plants still need to be explored further [28]. Thus, the effects of biochar, activated phosphate rock, and a combination of the two on plant growth and the uptake of nutrition by plants were investigated in this work.

The degree of improvement observed in the plant growth in the composite amendment treatments was obviously higher than the application of biochar and activated phosphate rock individually. Especially during the second crop planation, the uptake of P, Ca, and Mg by the two kinds of vegetables was significantly enhanced by the application of BCAPR. The soil in the BCAPR treatments had the highest contents of available nutrients in all the amendment treatments, which also supported the above results. Thus, increased plant heights in the pepper plants grown with the composite amendments were significantly higher than that of water spinach. The application of biochar functionalized with layered double hydroxides to slowly supply phosphorus was observed by Buates et al. [32]. For the variations in the available nutrition concentration in the sandy soil samples, the amendment treatments containing phosphate rock and activated phosphate rock showed a higher available P contents than that of the other amendments (Table 1). The above results indicated that the composite amendments were able to continuously improve the nutrient supply. This may be due to the reduction in P release kinetics induced by the formation of a biochar barrier and the contact between P and the soil in the fertosphere [33]. The FTIR results suggested that the reduction in aromaticity for both the BCPR and BCAPR amendments compared to biochar, which might be caused by the slowdown of the carbonization rates induced by phosphate rock [34]. Compared to biochar, the signals were strengthened in the spectra of the activated phosphate rock, BCPR and BCAPR, suggesting the conversion of insoluble Ca-P [12]. Moreover, the peaks around 1094 and 1046 cm^−1^ with the characteristic of P-O-C and P-O-P stretching were the main forms of phosphate groups in the activated phosphate rock, BCPR and BCAPR. As a high-efficiency adsorbent, biochar could absorb the released P, preventing its conversion to unavailable forms. It was also demonstrated that biochar could improve the availability of P via sorption from fertilizers and reduce the P leaching, which resulted in a slowdown in the release of P from the adsorbent [35,36]. Furthermore, it was easier for the chelated P to be retained by BC than soluble P.

### 4.2. The Synergistic Mechanisms of Biochar and Activated Phosphate Rock on Cd and Pb Immobilization

Previous studies revealed that amendments had a heavy immobilization effect on toxic metals depending on the types of metal, for example, oxalic acid-activated phosphate rock was more effective than bone meal for the immobilization of Pb in soil contaminated with Cu and Pb, while bone meal was able to immobilize Cu more efficiently than oxalic acid-activated phosphate rock [11]. For the formation of metal-P complex in contaminated soil, the type of P amendment is a crucial factor [37]. In this study, the biochar combined with activated phosphate rock was the most efficient amendment for both Cd and Pb immobilization, which was related to the enhanced SOM and pH, as well as the supply of available P (Table 1). The soil in the BCAPR treatments had the lowest contents of Cd and Pb in all the amendment treatments, which provided evidence for the above results. Given the unique capacity of biochar, to improve the soil organic carbon pool, water-holding capacity, and the nutrient supply capacity, the contents of soil organic matter were significant elevated in the amendment treatments that contained biochar compared to the other amendments. The porous structure and rough surface of the activated phosphate rock were observed by SEM, and their aggregated porous structure are closely related to the high sorption ability of heavy metals [28]. The FTIR results indicated that the presence of phosphate in the biochar matrix reduced the mobility of Pb through the formation of insoluble mineral precipitates of Pb_5_(PO_4_)_3_OH and Pb_5_(PO_4_)_3_Cl [38,39]. Luo et al. also reported that the formation of stable precipitates Pb_5_(PO_4_)_3_Cl and Cd_3_(PO_4_)_2_ resulted in biochar combined with Ca(H_2_PO_4_)_2_ or Ca_5_(PO_4_)_3_(OH) having higher Pb/Cd immobilization (31.3–92.3%) compared to pristine biochar (9.5–47.2%) [40]. Furthermore, the application of humic acid was beneficial for heavy metal immobilization, which could be ascribed to the high absorbing capacity of activated phosphate rock (Appendix A).

## 5. Conclusions

The synergistic effect of the composite amendment made with activated phosphate rock and biochar on the nutrition supply and immobilization of toxic metals in the tropical sandy soil was verified in this study. Comparing to the single amendments, the composite amendment BCAPR was the most effective for increasing the Ca, Mg, and P uptake by vegetables. The activation process by humic acid induced the change from insoluble or citrate-soluble Ca-P into highly active HPO_4_^2−^, thereby increasing the release of P, Ca, and Mg from phosphate rocks and increasing their availability in plants. Additionally, biochar played an important role in the retention of the available nutrition elements. Among the five amendments, BCAPR amendment showed the highest immobilization effectiveness for both Cd and Pb. It was closely related to the increased soil organic matter and the formation of insoluble mineral precipitates. In the practical application of biochar as a soil amendment, the combination of activated phosphate rock and biochar is beneficial when it comes to increasing nutrition supply and immobilizing the potential toxic elements in tropical sandy soils. The micro-mechanisms of the synergistic effect of the composite amendment of activated phosphate rock and biochar in tropical sandy soil need to be further explored, as do the long-term effects of its application.

## Figures and Tables

**Figure 1 ijerph-19-06431-f001:**
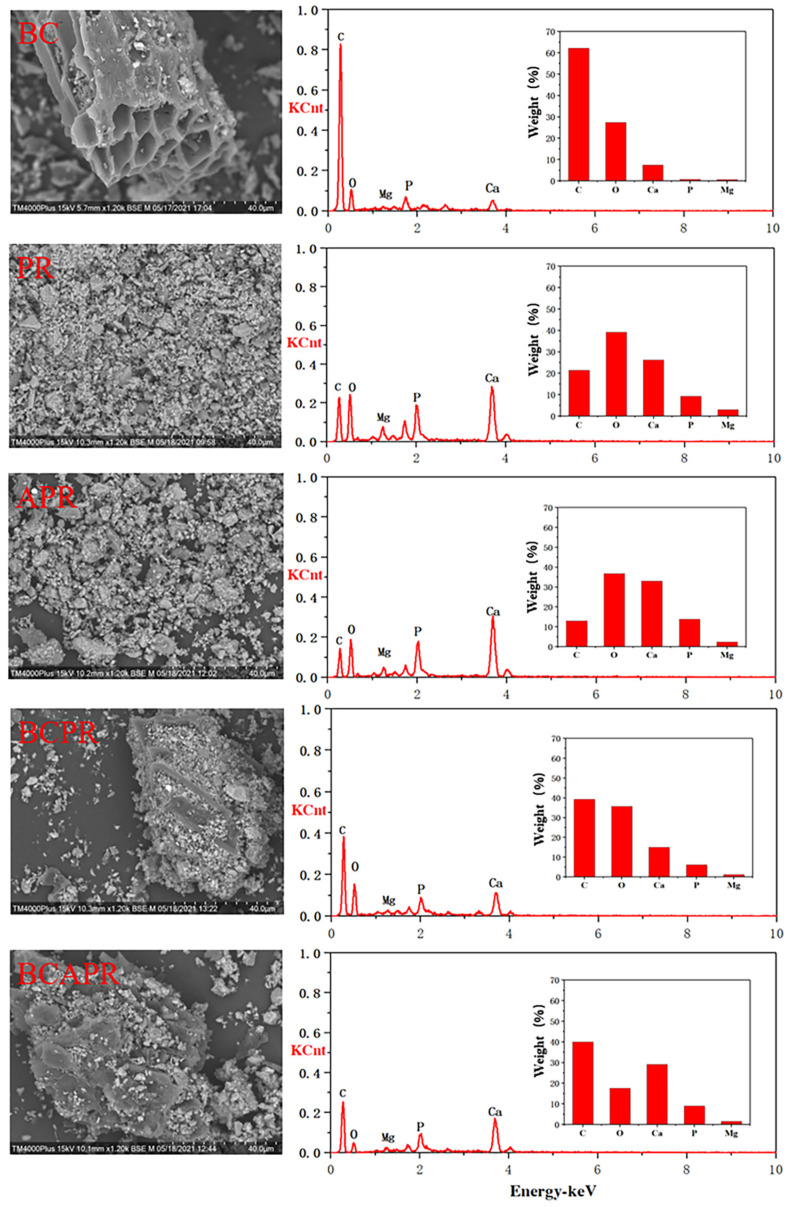
SEM images and the corresponding EDS spectra of the different amendments. The number of replicates was three. KCnt represented 1000 counts. BC: biochar, PR: phosphate rock, APR: activated phosphate rock, BCPR: composite of biochar and phosphate rock, BCAPR: composite of biochar and activated phosphate rock.

**Figure 2 ijerph-19-06431-f002:**
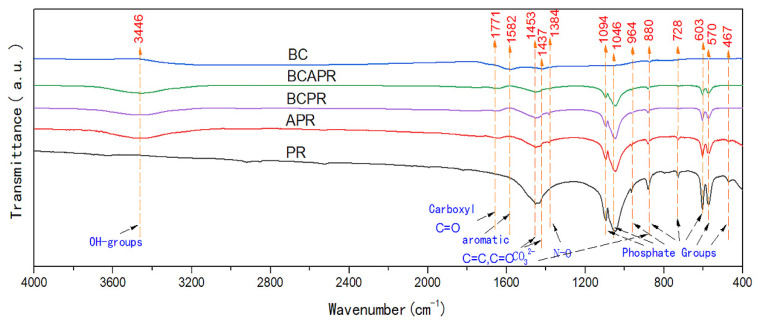
FTIR spectra of the different amendments. The number of replicates was three. BC: biochar, PR: phosphate rock, APR: activated phosphate rock, BCPR: composite of biochar and phosphate rock, BCAPR: composite of biochar and activated phosphate rock.

**Figure 3 ijerph-19-06431-f003:**
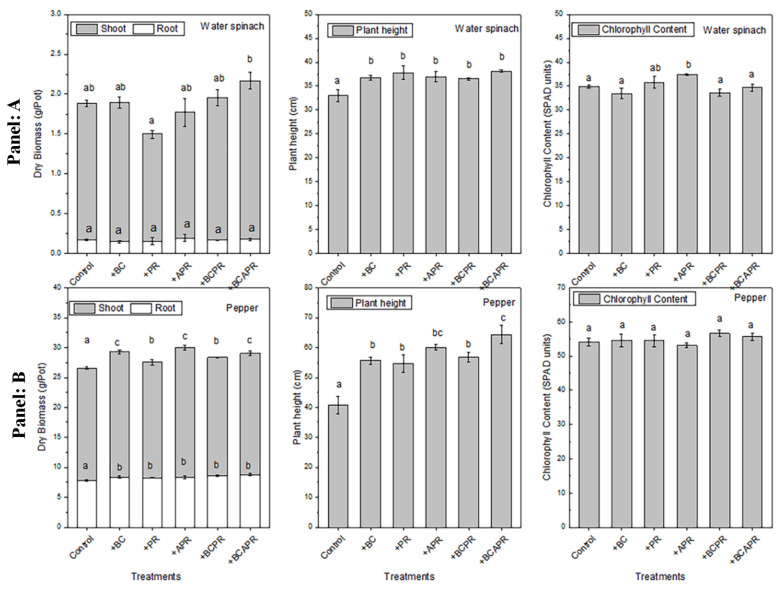
Plant heights, leaf chlorophyll contents and biomass of water spinach and pepper plants in successive cultivations with different amendments. The number of replicates was three. Different lowercase letters indicate significant differences at *p* < 0.05. Panel A: first cultivation, Panel B: second cultivation, BC: biochar, PR: phosphate rock, APR: activated phosphate rock, BCPR: composite of biochar and phosphate rock, BCAPR: composite of biochar and activated phosphate rock.

**Figure 4 ijerph-19-06431-f004:**
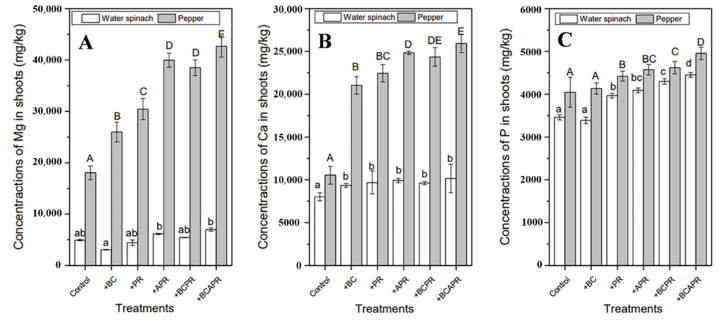
Mg (**A**), Ca (**B**), and P (**C**) uptake by water spinach and pepper with different amendments. The number of replicates was three. Different lowercase letters indicate significant differences in water spinach at *p* < 0.05. Different capital letters indicate significant differences in pepper at *p* < 0.05. BC: biochar, PR: phosphate rock, APR: activated phosphate rock, BCPR: composite of biochar and phosphate rock, BCAPR: composite of biochar and activated phosphate rock.

**Figure 5 ijerph-19-06431-f005:**
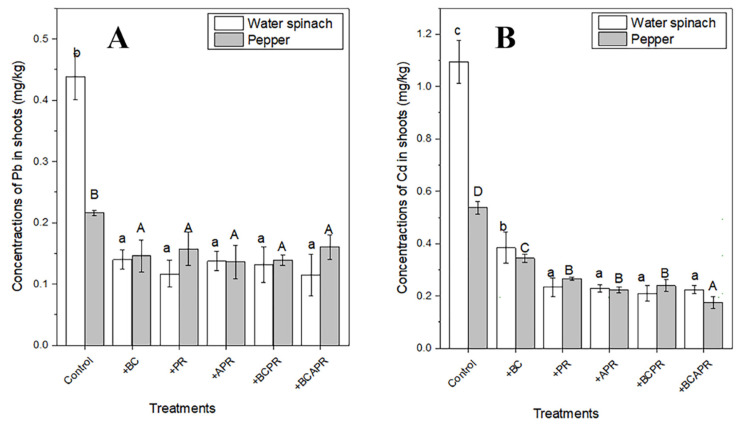
Pb (**A**) and Cd (**B**) uptake by water spinach and pepper in different amendment treatments. The number of replicates was three. Different lowercase letters indicate significant differences in water spinach at *p* < 0.05. Different capital letters indicate significant differences in pepper at *p* < 0.05. BC: biochar, PR: phosphate rock, APR: activated phosphate rock, BCPR: composite of biochar and phosphate rock, BCAPR: composite of biochar and activated phosphate rock.

**Table 1 ijerph-19-06431-t001:** Soil characteristics with different treatments.

	pH	Organic Matter (mg/kg)	Available P (mg/kg)	Available Ca (mg/kg)	Available Mg (mg/kg)	Extractable Cd (μg/kg)	Extractable Pb (μg/kg)
				**First crop**			
**Control**	5.54 ± 0.05 ^a^	8.14 ± 0.56 ^a^	66.31 ± 4.46 ^a^	6.22 ± 1.02 ^a^	23.51 ± 0.47 ^a^	10.10 ± 0.15 ^d^	33.20 ± 0.64 ^b^
**+BC**	6.95 ± 0.05 ^c^	9.73 ± 0.12 ^b^	56.48 ± 5.22 ^a^	287.21 ± 15.61 ^b^	31.22 ± 2.62 ^a^	3.02 ± 0.48 ^c^	23.10 ± 1.12 ^a^
**+PR**	7.60 ± 0.11 ^d^	8.44 ± 0.14 ^ab^	103.26 ± 3.75 ^b^	385.27 ± 3.55 ^c^	102.27 ± 3.78 ^b^	2.31 ± 0.09 ^bc^	26.90 ± 0.54 ^b^
**+APR**	6.71 ± 0.09 ^b^	9.03 ± 0.20 ^abc^	103.21 ± 4.53 ^b^	447.34 ± 15.21 ^d^	160.09 ± 13.31 ^c^	1.46 ± 0.20 ^a^	23.00 ± 0.19 ^a^
**+BCPR**	7.65 ± 0.02 ^d^	9.63 ± 0.39 ^bc^	112.17 ± 1.31 ^b^	452.08 ± 19.48 ^de^	187.24 ± 4.88 ^d^	1.73 ± 0.28 ^ab^	22.60 ± 1.09 ^a^
**+BCAPR**	6.78 ± 0.01 ^bc^	10.30 ± 0.87 ^c^	124.09 ± 1.62 ^c^	490.17 ± 12.07 ^e^	219.33 ± 7.06 ^e^	1.35 ± 0.06 ^a^	20.20 ± 1.38 ^a^
				**Second crop**			
**Control**	4.81 ± 0.10 ^a^	8.54 ± 0.07 ^b^	68.62 ± 4.64 ^a^	8.89 ± 0.54 ^a^	25.32 ± 0.47 ^a^	5.31 ± 0.34 ^d^	34.50 ± 1.89 ^b^
**+BC**	7.34 ± 0.03 ^d^	10.10 ± 0.22 ^c^	70.08 ± 3.85 ^a^	329.15 ± 27.12 ^b^	75.69 ± 6.25 ^b^	4.04 ± 0.41 ^c^	29.10 ± 1.82 ^a^
**+PR**	7.59 ± 0.09 ^e^	7.45 ± 0.20 ^a^	121.12 ± 4.61 ^b^	358.06 ± 22.58 ^b^	125.43 ± 11.21 ^c^	3.49 ± 0.17 ^bc^	27.90 ± 1.59 ^a^
**+APR**	6.29 ± 0.04 ^b^	8.87 ± 0.09 ^b^	116.31 ± 4.15 ^b^	733.32 ± 62.43 ^c^	192.26 ± 2.54 ^d^	2.69 ± 0.42 ^b^	25.90 ± 0.71 ^a^
**+BCPR**	7.78 ± 0.01 ^f^	9.09 ± 0.20 ^b^	121.24 ± 0.73 ^b^	808.11 ± 5.62 ^cd^	229.12 ± 3.51 ^e^	1.48 ± 0.10 ^a^	26.08 ± 1.09 ^a^
**+BCAPR**	6.64 ± 0.03 ^c^	10.10 ± 0.62 ^c^	147.32 ± 22.31 ^b^	875.24 ± 7.18 ^d^	256.34 ± 4.69 ^f^	0.86 ± 0.06 ^a^	25.80 ± 1.03 ^a^

Different letters in each column indicate significant differences between different treatments application (*p* < 0.05; Duncan’s test). The number of replicates was three. BC: biochar, PR: phosphate rock, APR: activated phosphate rock, BCPR: composite of biochar and phosphate rock, BCAPR: composite of biochar and activated phosphate rock.

## Data Availability

All data generated or analyzed during this study are included in this published article and its Appendix A.

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
