# Peer review of "The Synergistic Effect of Biochar-Combined Activated Phosphate Rock Treatments in Typical Vegetables in Tropical Sandy Soil: Results from Nutrition Supply and the Immobilization of Toxic Metals"

_ijerph, 2022, doi:10.3390/ijerph19116431_

Round 1
Reviewer 1 Report
The English is generally good; well enough to be understood. However, a through review needs to be conducted prior to acceptance for proper English syntax. There are numerous syntax, grammar or punctuation errors; for example:
line 19 - effects are still need to further explore.
line 20 - pot experiment was conducted
similar errors on lines 67, 78, 82, 87, 90, 93, 185, 189, etc.
The experiments are described clearly. The results are significant as evident from Figures 1 and 2. The interpretation of results is sound.
CHANGES NEEDED:
- Figure 1 - There needs to be some identification of the 6 graphs. I suggest labeling the top 3 "Panel A" and the lower 3 "Panel B". Then in the Figure legend line2 201-204 there needs to be a corresponding clarification "Panel A - First Cultivation"; Panel B - Second Cultivation.
- Figure 2 - also needs to have the three graphs in the Figure labeled "A. B. and C." Then in the figure legend line 219 there needs to be a corresponding clarification "Panel A - Mg treatment; Panel B - Ca treatment; Panel C - P treatment"
- line 51 - clarify: What is "water-soluble superphosphate"???
Reviewer 2 Report
The manuscript aimed to investigate the synergistic effect of biochar combined activated phosphate rock on immobilization of toxic metals and the nutrition supply of typical vegetables in tropical sandy soil. Some interesting data were presented by the study, however, major revisions are required in some parts. Moreover, there exist some speculations in the discussion of results, and the conclusions are not convincible without supporting by solid experimental evidences. See below:
- Line 145: How to understand?
- Line 195-198: There was no obviously increment of leaf chlorophyll content in water spinach under phosphate rock, activated phosphate rock and BCAPR treatments.
- Line 272-274: “activated phosphate rock that fabricated by humic acid showed a higher capacity of toxic metal passivation for both water spinach and pepper” is inaccurate presentation.
- Line 289-293: Just inference, lack of data support.
- Line 303-308: It is not easy to find the result as your description, BET will be more intuitive.
- Line 330-331: Compared to biochar, the signals were strengthened in the spectra of activated phosphate rock, BCPR and BCAPR.
- Line 331-338: Speculation seems unreasonable.
- Line 353-356: There is no data support, only speculation.
Reviewer 3 Report
The authors present some interesting data on the application of biochar and phosphate rock in sandy soils which may well be suitable for publication. However, there are some issues with the manuscript which need to be improved significantly before it can finally be considered for publication. First, the language quality is insufficient and needs to be largely improved, starting with the title. Several sentences, especially in the introduction and the discussion, are hardly understandable which makes it difficult to evaluate the respective content. I highly suggest using a professional English editing service for this purpose. Second, the discussion is too short and does not discuss many of the shown parameters, e.g. plant characteristics shown in the results or differences between spinach and pepper. The authors need to either reduce the data shown in the manuscript so that it is more focused or, which would be the better way, expand the discussion to really discuss all results shown and to create an actual synthesis of the results which would be of higher interest to the reader than the current discussion. See below for more specific comments.
Line 103-105: There needs to be more information on the sites and the soil, e.g. pH, soil carbon as well as soil type, land use etc.
Line 105-108: This needs to be stated in the introduction and not in the M&M section.
Line 110-113: There needs to be more information on properties of the applied biochar and phosphate rock.
Line 138: The height of the pot is missing and the resulting bulk density of the soil.
Line 139: What do you mean by aging and equilibrium?
Line 172: Did you check for prerequisites for these analyses? Please describe here. Also, please add for which parameters you performed which analyses.
Results section: In general, you have very clear results which make it easy to get the general direction of your findings, so you do not need to describe your results in such detail. Please shorten the results section.
Line 196-199: From the figure, these differences are not significant, and anyway not large. Please adjust these sentences.
Figure 1: There seems to be a mistake with the letters concerning water spinach and chlorophyll – you show “ab” as letters, but there is no single “b”.
Line 200: All figure descriptions should contain information on the number of replicates and the variance estimate shown, as well as on the letters indicating significant differences.
Line 207-208: This is not true for all combinations of nutrients and plants – please correct.
Table 1: This table needs to be completely revised. Right now, it is hard for the reader to get anything from it. The table needs to much bigger to be readable at all. Additionally, the authors should think about reducing it or changing the layout so that the relevant results really become clear for the reader without searching for them. Right now, it is very confusing and chaotic.
Discussion: I do not think the structure of your discussion is right. You first discuss the effect of the amendments on plant growth, then you talk about properties of the amendments themselves. It would make much more sense to do it the other way around, also in the results section. Please restructure the results and the discussion.
Line 302: You cannot present new data in the discussion. Please present the data for itself in the results section and then discuss it in the discussion.
Line 309-310: How do you get to these numbers?
Figure 4: The y axes in the figure need to have the same range, otherwise the presentation of the data is misleading. Also, you need to explain the unit of the y axes.
Line 318: The same applies here – you need to present the data in the results section and then only discuss it in the discussion section.
Line 318-338: The largest part of this section belongs to the results as it merely describes the data. Please transfer to the right section and use the discussion section for an actual discussion of the data.
Conclusion: The conclusion is too much of a summary. Please concentrate on the core of your results and the importance for the practice as well as further research needs.
Round 2
Reviewer 3 Report
The authors have improved the manuscript in some parts, but some issues remain.
There are still many typos and formatting issues in the manuscript so that I honestly find it hard to believe that an editing service saw this version of the manuscript. This also leads to the impression that the authors revised the manuscript sloppily. Please check the whole manuscript so that there are no typos and weird formatting issues in the text anymore. Overall, the quality of the language is still not high concerning grammatical errors, word choice and syntax. Again, using the help of native speakers is strongly encouraged before the manuscript can be considered for publication.
Contrary to the statement of the authors, I did not find added information on “aging and equilibrium” in the manuscript. Which and how the prerequisites for the statistical analyses were checked should also be described in detail in the manuscript.
The authors added details on the pot dimension and the bulk density of the soil, as suggested. But why was the extremely low bulk density 0.48 g/cm³ chosen for this experiment? This certainly does not reflect the bulk density of any naturally occurring mineral soil, so this questions the whole experiment. The authors need to justify this choice in detail in the manuscript and describe why they think the results of the study are still meaningful. This is crucial for the evaluation of the study.
The table was improved, however, it is still quite unclear. It would be easier if the treatments and parameters were exchanged so that the differences between the treatments would be visible within one column instead of one line. There should also be a uniform number of positions behind the decimal point, at least for one parameter. If the parameters would be presented in one column again, the numbers should be aligned at the decimal points. Also, the table description needs to be much more detailed concerning the data you present (mean of x replicates, standard deviations? and information on the letters). If line 306ff is meant to be a footnote to the table, this needs to be formatted differently.
There is still no uniform range of the y axes in figure 1 and no description of the axes’ unit despite the authors stating that they changed this in the reply. The headline of 4.1 does not fit to the content of the following text. The conclusion is still not enough of a conclusion, the relevance of the results for practical implementation should be more clearly stated instead of summarizing the results.
